# TCF12 and LncRNA MALAT1 Cooperatively Harness High Cyclin D1 but Low β-Catenin Gene Expression to Exacerbate Colorectal Cancer Prognosis Independently of Metastasis

**DOI:** 10.3390/cells13242035

**Published:** 2024-12-10

**Authors:** Chia-Ming Wu, Chung-Hsing Chen, Kuo-Wang Tsai, Mei-Chen Tan, Fang-Yu Tsai, Shih-Sheng Jiang, Shang-Hung Chen, Wei-Shone Chen, Horng-Dar Wang, Tze-Sing Huang

**Affiliations:** 1Graduate Program of Biotechnology in Medicine, National Health Research Institutes, National Tsing Hua University, Hsinchu City 300, Taiwan; herroyuihk@gmail.com; 2Institute of Biotechnology, National Tsing Hua University, Hsinchu City 300, Taiwan; 3National Institute of Cancer Research, National Health Research Institutes, Miaoli County 350, Taiwan; chchen@utaipei.edu.tw (C.-H.C.); maggietan@nhri.edu.tw (M.-C.T.); tatufish@nhri.edu.tw (F.-Y.T.); ssjiang@nhri.edu.tw (S.-S.J.); bryanchen@nhri.edu.tw (S.-H.C.); 4Department of Mathematics, University of Taipei, Taipei City 100, Taiwan; 5Department of Research, Taipei Tzu Chi Hospital, Buddhist Tzu Chi Medical Foundation, New Taipei City 231, Taiwan; kwtsai6733@gmail.com; 6Division of Colon & Rectal Surgery, Department of Surgery, Taipei Veterans General Hospital, Taipei City 112, Taiwan; wschen@vghtpe.gov.tw; 7Department of Biochemistry, School of Medicine, Kaohsiung Medical University, Kaohsiung City 807, Taiwan; 8Doctoral Program in Tissue Engineering and Regenerative Medicine, Biotechnology Center, National Chung Hsing University, Taichung City 402, Taiwan

**Keywords:** TCF12, MALAT1, WNT pathway, β-catenin, cyclin D1

## Abstract

Metastasis is a well-known factor worsening colorectal cancer (CRC) prognosis, but mortality mechanisms in non-metastatic patients with poor outcomes are less understood. TCF12 is a transcription factor that can be physically associated with the long non-coding RNA MALAT1, creating an alliance with correlated expression levels in CRC patients. This TCF12–MALAT1 alliance is linked to poorer prognosis independently of age and metastasis. To identify the downstream effects responsible for this outcome, we analyzed 2312 common target genes of TCF12 and MALAT1, finding involvement in pathways like Aurora B, ATM, PLK1, and non-canonical WNT. We investigated the impact of WNT downstream genes *CTNNB1* and *CCND1*, encoding β-catenin and cyclin D1, respectively, on survival in CRC patients with this alliance. Tumors with higher *TCF12* and *MALAT1* gene expressions alongside increased *β-catenin* gene expressions were classified as having a “Pan-CMS-2 pattern”, showing relatively better prognoses. Conversely, tumors with high *TCF12*, *MALAT1*, and *cyclin D1* gene expressions but low *β-catenin* expression were categorized as “TMBC pattern”, associated with poor survival, with survival rates dropping sharply from 60% at one year to 30% at three years. This suggests that targeting cyclin D1-associated CDK4/6 could potentially reduce early mortality risks in TMBC patients, supporting personalized medicine approaches.

## 1. Introduction

TCF12 (also called HEB or HTF4) is a class-I member of the helix–loop–helix (HLH) protein family and functions as a transcription factor that directly binds to the E-box sites (CANNTG) of target gene promoters [1,2]. There are 15,373 genes identified as TCF12 targets based on the ENCODE Transcription Factor Targets Dataset using DNA sequencing analyses following anti-TCF12 chromatin immunoprecipitation (ChIP) [3,4]. The fact that TCF12 has such a vast number of target genes suggests that it can be a highly influential transcription factor. Indeed, TCF12 regulates a wide range of cellular processes, e.g., the mesenchymal transition of epithelial/endothelial cells [5,6,7], fibroblast activation and extracellular matrix remodeling [8], the drug uptake of cancer cells [9], and muscle development [10]. In order for TCF12 to correctly regulate the expression of the proper downstream genes in the responsible cells under the right conditions, it likely needs to cooperate closely with other regulatory proteins and RNAs to ensure that it plays an accurate role in the transcriptional machinery complex. For example, TCF12 heterodimerizes with TWIST1, a class-II member of the HLH family, which is required for the proper formation of the junction structure of the frontal and parietal bones during development [11]. The heterodimer formed by TCF12 and MyoD, another class-II member of the HLH family, is involved in myogenesis [10]. When TCF12 forms a complex with Id-1, a class-V member of the HLH family, TCF12 is unable to bind to the promoters of target genes for transcriptional regulation [9,12]. Additionally, TCF12 can function as a transcriptional repressor by associating with Bmi1 and EZH2, the components of polycomb-repressive complex 1 (PRC1) and PRC2, respectively [5]. Apart from the aforementioned regulatory proteins, the long non-coding RNA (lncRNA) MIAT can also form a complex with TCF12, promoting its binding to the *NFAT5* gene promoter to activate gene expression [13]. TCF12 overexpression has been detected in many malignancies such as colorectal cancer [5], breast cancer [8], gallbladder cancer [14], hepatocellular carcinoma [15], melanoma [13], and pancreatic cancer [16].

While ~80% of the human genome can be transcribed into RNA, only ~1.5% of whole genomic DNA is responsible for the synthesis of protein-coding transcripts [17,18]. Most of the human genome expresses non-coding RNA, which is involved in diverse biological processes, including the regulation of mRNA expression. Non-coding RNA with a length >200 nucleotides is generally classified as lncRNA, distinguishing it from small-sized non-coding RNA, including microRNA (miRNA), Piwi-interacting RNA (piRNA), and small nucleolar RNA (snoRNA), among others. The first identified and widely studied lncRNA is Metastasis-Associated Lung Adenocarcinoma Transcript 1 (MALAT1), which is also named Nuclear-Enriched Abundant Transcript 2 (NEAT2) [19]. MALAT1 has a length of 8664 nucleotides and exerts its versatile functions by interacting with many diverse factors. For example, it is involved in the alternative processing of pre-mRNA by recruiting and modulating splicing-regulatory proteins into nuclear speckles [20,21,22]. MALAT1 participates in a transcription-repressive complex by associating with PRC2 components EZH2, SUZ12, and EED to drive the trimethylation of histone H3 at lysine-27 and the repression of target gene promoters [23,24], but it acts as a molecular scaffold promoting the interactions among unmethylated Pc2, E2F, and other transcription factors to trans-activate downstream gene expression [25]. It can also play a role of guiding RNA to facilitate Sp1 binding to the target gene promoter [26]. Additionally, MALAT1 can function as a competing endogenous RNA to sponge miRNA and in turn up-regulate the miRNA targets [27,28,29]. MALAT1 was discovered because its high expression was associated with tumor metastasis and patients’ poorer survival in early-stage non-small-cell lung cancer [19]. Its abnormally elevated expression has been subsequently observed in various cancers such as hepatocellular carcinoma [30], breast cancer [31], colorectal cancer [32], prostate cancer [33], gastric adenocarcinoma [34], multiple myeloma [35], and esophageal cancer [36]. MALAT1 overexpression exhibits tumor-promoting effects through the promotion of tumor cell growth, proliferation, migration, invasion, metastasis, and survival [18].

Despite the reported clinical relevance of TCF12 and MALAT1 in CRC malignancy [5,32], their individual overexpression in the patient cohort from The Cancer Genome Atlas—Colon Cancer (TCGA-COAD) dataset does not show a significant association with poorer prognosis. Recognizing the multifaceted roles of TCF12 and MALAT1 through interactions with various factors, we employed RNA immunoprecipitation and sequencing to investigate the interactions between TCF12 and lncRNAs. Our findings revealed that MALAT1 is physically associated with TCF12 in CRC cells, and their combined expression was identified as a metastasis-independent prognostic factor for reduced overall survival in CRC patients. We term this relationship between TCF12 and MALAT1 the TCF12–MALAT1 alliance. To uncover downstream factors/events of the alliance contributing to this poorer prognosis, we performed signal pathway enrichment analysis on their shared 2312 target genes, identifying pathways such as Aurora B, ATM, PLK1, and non-canonical WNT as significantly involved. Moreover, through the TCGA-COAD Genome Analyzer (GA) dataset, we identified two distinct CRC patient subgroups: the “Pan-CMS-2 pattern”, associated with better survival, and the “TMBC pattern”, linked to poorer relapse-free and overall survival regardless of metastasis. Our findings provide support for developing personalized medicine to potentially reduce the risk of early mortality in CRC patients.

## 2. Materials and Methods

### 2.1. Cell Culture

Human CRC cell lines SW620 and SW480 were cultivated with RPMI-1640 medium supplemented with 10% of fetal bovine serum, 100 units/mL of penicillin, 100 μg/mL of streptomycin, and 2 mM of L-glutamine, which was conducted at 37 °C in a humidified incubator filled with 95% air and 5% CO_2_.

### 2.2. TCGA-COAD Dataset

We investigated the clinical implications of higher TCF12 mRNA expression, higher MALAT1 expression, and the TCF12–MALAT1 alliance in CRC using RNA sequencing and patients’ clinical data of the TCGA-COAD dataset acquired from the TCGA Hub in the UCSC Xena website [37]. The RNA sequencing data, transformed into log2 (x + 1) RSEM normalized counts, were downloaded and used for subsequent analyses without any preprocessing. The clinical information, including patient’s gender, age, and pathologic stage, was downloaded from the Phenotypes file, and relapse-free survival (RFS) and overall survival (OS) were determined based on the curated survival data available through Xena. The tumor purity of the TCGA-COAD dataset was estimated using the ESTIMATE (Estimation of STromal and Immune cells in MAlignant Tumor tissues using Expression data) algorithm [38]. We used ssGSEA (single-sample Gene Set Enrichment Analysis) to calculate the stromal and immune scores based on 141 immune-related genes and 141 stromal-related genes and then predicted tumor purity using the combination of the two scores. Additionally, the patients of the TCGA-COAD dataset were classified into 4 subtypes based on the criteria of the Consensus Molecular Subtype (CMS) classification [39], including CMS-1 (i.e., microsatellite-unstable subtype), CMS-2 (i.e., canonical WNT subtype), CMS-3 (i.e., metabolism-dysregulated subtype), and CMS-4 (i.e., mesenchymal subtype).

### 2.3. RNA Immunoprecipitation (RNA IP)

RNA IP was performed to identify TCF12-associated lncRNAs in CRC cells [40]. SW620 cells were harvested using trypsinization and washed three times with ice-cold PBS. The cell suspension (5 × 10^7^ cells in 10 mL of PBS) was added with 162 μL of 18.5% formaldehyde for a cross-linking reaction at room temperature. After 15 min, the reaction was stopped by the addition of 1 mL of 2.5 M glycine and incubation for another 5 min. The cells were collected by centrifugation, washed twice with ice-cold PBS, and further mixed with 1 mL of lysis buffer (consisting of 50 mM of Tris-HCl, pH 7.4, 1% NP-40, 0.5% sodium deoxycholate, 0.1% SDS, 1 mM of EDTA, and 150 mM of NaCl) plus 1 mM of dithiothreitol, cocktails of protease inhibitors (Sigma-Aldrich, St. Louis, MO, USA), and 40 U/mL of RNasin (Invitrogen, Carlsbad, CA, USA) on ice for 10 min. The mixture was further homogenized by an ultrasonic homogenizer and then centrifuged at 12,000 rpm for 15 min at 4 °C. One-third of the supernatant was saved as the INPUT fraction. The remaining supernatant was divided into two aliquots for an overnight incubation at 4 °C with 5 μg of control IgG and anti-TCF12 antibody (Cat. No. sc-28364, Santa Cruz Biotechnology, Santa Cruz, CA, USA), respectively. Subsequently, the two aliquots were further incubated for 30 min at room temperature with 50 μL of protein G-conjugated magnetic beads. The beads were collected using a magnet and washed three times with lysis buffer. One half of the beads of each sample were resuspended in 100 μL of Laemmli buffer and heated at 95 °C for 15 min prior to 10% SDS-polyacrylamide gel electrophoresis, followed by conventional immunoblot analysis with anti-TCF12 antibody (1:500 dilution, Cat. No. sc-28364, Santa Cruz Biotechnology). Another half of the beads of each sample were resuspended in 100 μL of reversion buffer (consisting of 50 mM Tris-Cl, pH 7.4, 150 mM NaCl, 5 mM EDTA, 10 mM dithiothreitol, 1% SDS, and 10 μg of proteinase K) and incubated at 42 °C for 30 min and 65 °C for 4 h before RNA isolation by TRIsure–chloroform extraction and RNeasy Mini column chromatography (Qiagen, Germantown, MD, USA). The TCF12-associated MALAT1 was detected by PCR. The primer sequences were as follows: forward, 5′-TCA-TAC-CTA-ACC-AGG-CAT-AAC-A-3′; reverse, 5′-AAG-TGC-TCA-CAA-GGC-AAA-TC-3′. The PCR condition was 95 °C (30 s), 56 °C (40 s), and 72 °C (40 s) for 30 cycles. The size of the PCR product was 206 bp.

### 2.4. RNA Library Sequencing

RNA samples, isolated from the aforementioned control IgG or anti-TCF12 immunoprecipitates and the INPUT control, were subjected to next-generation sequencing to identify TCF12-associated lncRNAs in CRC cells. First, RNA libraries were constructed according to the manufacturer’s protocol of the NEBNext^®^ Small RNA Library Prep Kit (New England Biolabs, Ipswich, MA, USA). The 5′ and 3′ adaptors were ligated to total RNA prior to subsequent reverse transcription and PCR amplification. The enriched cDNA fragments were size-fractionated by a 6% polyacrylamide gel electrophoresis, and the 100~200-nucleotide RNA fragments (140~155 nucleotides in length with both adapters) were purified for library construction. Small RNA libraries were subsequently sequenced using the Illumina MiSeq 150-cycle single-read platform (Illumina, San Diego, CA, USA). The generated sequence reads went through a quality control process to remove low-quality reads, followed by 3′ adaptor trimming to generate clean reads [41,42]. Only clean reads with read counts ≥ 2 were included for further analyses. They were aligned to UCSC human hg19 or the Ensembl database by the Bowtie 2 tool [43], and the significant peaks were called using the cisGenome tool [44]. The fold changes in the peaks were calculated by dividing the average fragments per kilobase per million (FPKM) across the RNA IP samples to the FPKM of the INPUT control. Transcripts were identified as TCF12-associated RNAs when their fold changes were >2-fold.

### 2.5. Gene Set Enrichment Analysis

The downstream target genes of TCF12 were obtained from the ENCODE dataset, and the target genes of MALAT1, predicted by the LncRNA2Target database, were searched from the UALCAN website of the University of Alabama at Birmingham [45]. We applied common target genes of TCF12 and MALAT1 to Model-based Gene Set Analysis (MGSA) for signal pathway enrichment analysis [46].

### 2.6. Statistical Analyses

The correlation between the expression levels of the two genes was analyzed using Pearson’s correlation analysis. Differences in the ages and gene expression levels between the two patient subsets were evaluated by an independent-samples *t*-test. Pearson’s Chi-square analysis was performed to examine the association of gene expression status with various clinicopathological characteristics. The OS and RFS rates of different subsets of CRC patients were calculated using the Kaplan–Meier method and compared by the log-rank test. Prognostic associations of the test variables were evaluated using the univariate and multivariate Cox regression analyses. The interaction term of TCF12 mRNA and MALAT1 expression levels, referred to as “TCF12 × MALAT1”, was also subjected to Cox regression analyses. The evaluation of TCF12 × MALAT1 involves more than simply multiplying their expression levels; it also requires adjusting with their individual effects. As such, “TCF12 × MALAT1” was represented as three covariates in a multivariate Cox regression analysis: TCF12 mRNA expression level, MALAT1 expression level, and the multiplication of TCF12 mRNA and MALAT1 expression levels. These statistical analyses were performed using SPSS 11.0 software (SPSS Inc., Chicago, IL, USA).

## 3. Results

### 3.1. The TCF12–MALAT1 Alliance Is Associated with Poor Prognosis in CRC Patients

TCF12 may exert its functions through interacting with other transcription-regulatory proteins and lncRNAs. Therefore, we wondered which lncRNAs could be associated with TCF12 to have some clinical impact on CRC patients. SW620 cells were used for anti-TCF12 RNA IP and next-generation sequencing experiments. Their lysate was prepared for reacting with anti-TCF12 antibody, and then the immunoprecipitates were divided into two aliquots for further extraction of the cellular proteins and RNAs, respectively. The protein fraction was analyzed by an immunoblot assay to confirm that TCF12 was specifically immunoprecipitated (the left panel in Figure 1A). The RNA fraction was for RNA sequencing and further analyses through the NCBI and Ensembl databases. Several transcripts were suggested to be TCF12-associated lncRNAs, and MALAT1 is one of the lncRNAs of top rank (the right panel in Figure 1A). To validate whether TCF12 was indeed associated with MALAT1, the lysate of SW620 cells was subjected to anti-TCF12 immunoprecipitation. RNA was extracted from the immunoprecipitates for further RT-PCR analysis of MALAT1 existence. The result showed that MALAT1 was indeed co-immunoprecipitated with TCF12 (Figure 1B). This association between TCF12 and MALAT1 can also be confirmed from another CRC cell line SW480 with ectopic TCF12 overexpression (Appendix A). Furthermore, we intended to investigate the clinical implication of the TCF12 and MALAT1 combination in CRC patients. The TCGA-COAD dataset includes 456 CRC patients’ data obtained from the GA or HiSeq platform. Concerning the batch effects that probably existed between the GA and HiSeq platforms, we estimated the tumor purities of the employed tumor tissues using the ESTIMATE algorithm. As shown in Figure 1C, the data from the GA platform exhibited a smaller variation than those from the HiSeq platform. The TCGA-COAD (GA) data were thus used for further analyses. Pearson’s correlation analysis firstly revealed a significant correlation between the levels of TCF12 mRNA and MALAT1 in the human CRC specimens of the TCGA-COAD (GA) dataset (r = 0.377, *p* < 0.001). A relatively higher average level of MALAT1 expression (12.31 ± 1.34 vs. 11.49 ± 1.43, *p* < 0.001) was detected in the CRC tissues exhibiting higher (i.e., >median vs. ≤median) TCF12 mRNA expression levels (Table 1), suggesting that TCF12 not only formed a complex with MALAT1 but that also its gene expression was coordinated with that of *MALAT1* in CRC patients. We call this relationship between TCF12 and MALAT1 the TCF12–MALAT1 alliance. This alliance not only indicates the physical association between TCF12 and MALAT1 but also includes the interaction between the expression levels of TCF12 mRNA and MALAT1, referred to as “TCF12 × MALAT1”. To comprehensively evaluate the clinical implication of the TCF12–MALAT1 alliance in CRC prognosis, we conducted univariate Cox regression analyses on variates including gender, age, metastasis, TCF12 mRNA expression level, MALAT1 expression level, and TCF12 × MALAT1. As shown in Figure 1D, TCF12 mRNA expression and MALAT1 expression under an alliance rather than them alone were significantly associated with the poorer OS of CRC patients. Under the TCF12–MALAT1 alliance, the coefficients for TCF12 mRNA expression and MALAT1 expression were 3.696 and 3.354, respectively, with corresponding *p*-values of 0.029 and 0.023. These associations remained statistically significant even after being adjusted for tumor metastasis and patients’ age, suggesting that TCF12 could collaborate with MALAT1 to exacerbate CRC patients’ OS independently of patients’ age and tumor metastasis (Figure 1E). Additionally, we also analyzed the association between the TCF12–MALAT1 alliance and poorer RFS outcome, finding a similar trend as with poorer OS outcome. However, the results did not reach statistical significance (Appendix A).

### 3.2. The TCF12–MALAT1 Alliance Is Associated with β-Catenin-Independent Cyclin D1 Gene Expression

To investigate why the TCF12–MALAT1 alliance is significantly correlated with the poorer prognosis of CRC patients, we identified 2312 overlapping genes from the 15,373 downstream target genes of TCF12 and the 2650 downstream target genes of MALAT1 (Figure 2A). Through MGSA signal pathway enrichment analysis, it was suggested that these overlapping genes are involved in pathways such as Aurora B, ATM, PLK1, and non-canonical WNT pathways (Figure 2B). Our present study further addressed the relationships between the TCF12–MALAT1 alliance and WNT downstream genes *CTNNB1*, *CCND1*, and *MYC* that encode β-catenin, cyclin D1, and c-Myc, respectively. WNT2B is the only WNT ligand revealed on the common target gene list of TCF12 and MALAT1, which can induce both canonical and non-canonical WNT pathways. The expression of *cyclin D1* and *c-myc* genes can be induced through the canonical WNT–β-catenin pathway or mediated by the TCF12–MALAT1 alliance without the involvement of β-catenin. In the TCGA-COAD (GA) dataset, which includes 193 patients, the TCF12 mRNA levels showed significant positive correlations with the MALAT1, β-catenin mRNA, and cyclin D1 mRNA levels. However, the MALAT1 levels were only positively correlated with the TCF12 mRNA and cyclin D1 mRNA levels. The β-catenin mRNA levels, while positively correlated with the TCF12 mRNA levels, showed no significant correlation with the levels of MALAT1, cyclin D1 mRNA, and c-Myc mRNA (Figure 2C). Considering tumor heterogeneity occurring among patients, we tried to classify the 193 CRC patients of the TCGA-COAD (GA) dataset based on the criteria of the CMS classification. All of the 184 patients could be successfully assigned to CMS-1 (*n* = 32, 17.4%), CMS-2 (*n* = 70, 38.0%), CMS-3 (*n* = 29, 15.8%), and CMS-4 (*n* = 53, 28.8%), respectively (Figure 2C). The significant positive Pearson correlations among the TCF12 mRNA, MALAT1, β-catenin mRNA, and cyclin D1 mRNA levels were observed in the patients of CMS-2. For CMS-1 patients, the TCF12 mRNA levels were positively correlated with the levels of MALAT1 and β-catenin mRNA but not with the levels of cyclin D1 mRNA. Their MALAT1 levels were significantly correlated with the TCF12 mRNA levels but not with the β-catenin, cyclin D1, or c-Myc mRNA levels. In CMS-3 patients, TCF12 mRNA expression was correlated with cyclin D1 mRNA but not with MALAT1 or β-catenin mRNA expression. Their MALAT1 levels were not correlated with the TCF12, β-catenin, or cyclin D1 mRNA levels. The tumors in CMS-4 patients showed that the TCF12 mRNA levels were not related to the β-catenin, cyclin D1, or c-Myc mRNA levels, but their MALAT1 levels were significantly correlated with the TCF12 and cyclin D1 mRNA levels. These data together pointed out the existence of complicated relationships among the expression levels of *TCF12*, *MALAT1*, *β-catenin*, *cyclin D1*, and *c-myc* genes in CRC. Nevertheless, CMS-2 patients had a relatively better prognosis among all CRC patients (Figure 2D). The CRC specimens with higher *TCF12* and *MALAT1* gene expressions tended to be CMS-2-subtype tumors only when they also exhibited higher *β-catenin* gene expression or lower *cyclin D1* gene expression (Figure 2E). We, furthermore, investigated the cyclin D1 and β-catenin mRNA levels in CRC tissues that had higher expression levels of both TCF12 mRNA and MALAT1 (designated as the TCF12^hi^MALAT1^hi^ expression status in Table 2). Compared to CRC tissues without simultaneous higher expressions of TCF12 mRNA and MALAT1, the TCF12^hi^MALAT1^hi^ CRC tissues were detected to have a statistically significantly higher expression of cyclin D1 mRNA (12.27 ± 0.68 vs. 11.93 ± 0.60, *p* = 0.001). However, the difference between their β-catenin mRNA expression levels did not reach statistical significance (13.69 ± 0.45 vs. 13.56 ± 0.56, *p* = 0.108). These data suggest that β-catenin-independent higher *cyclin D1* gene expression can be a tumor-promoting factor responsible for the significant association of TCF12^hi^MALAT1^hi^ status with poor prognosis in CRC patients.

### 3.3. The TCF12–MALAT1 Alliance Exacerbates CRC Through Low β-Catenin but High Cyclin D1 Gene Expression

Furthermore, we investigated the impact of the *β-catenin* and *cyclin D1* gene expression levels on the overall survival of CRC patients with TCF12^hi^MALAT1^hi^ status. First, among the 193 patients in the TCGA-COAD (GA) dataset, 61 patients exhibited a TCF12^hi^MALAT1^hi^ expression status. When these patients were further divided into two groups based on whether their β-catenin mRNA levels were greater than the median, we found that patients with higher β-catenin mRNA expression had a better prognosis than those with a lower expression (*p* = 0.004, Figure 3A). Among the 193 patients, only 28 had the TCF12^hi^MALAT1^hi^β-catenin^lo^ expression status and showed a statistically significant poorer prognosis (*p* = 0.017, Figure 3B), but the Chi-square analysis showed no difference in tumor recurrence and metastasis (*p* = 0.728 and 0.918, respectively). When the 61 patients with the TCF12^hi^MALAT1^hi^ expression status were divided into two groups based on whether their cyclin D1 mRNA levels were greater than the median, we found that patients with higher cyclin D1 mRNA expression had poorer prognosis compared to those with lower expression (*p* = 0.041, Figure 3C). Among all 193 patients, 37 had the TCF12^hi^MALAT1^hi^cyclin D1^hi^ expression status, which was significantly associated with patients’ tumor recurrence (*p* = 0.018) but not metastasis (*p* = 0.515). Their poorer prognosis did not reach statistical significance (*p* = 0.147, Figure 3D). Next, we evaluated the combined effects of *β-catenin* and *cyclin D1* gene expression levels on the OS of CRC patients with the TCF12^hi^MALAT1^hi^ status. As shown in Figure 4A, in the TCGA-COAD (GA) dataset, there were 33 patients whose tumor tissues simultaneously expressed higher levels of TCF12 mRNA, MALAT1, and β-catenin mRNA. These patients showed no significant difference in OS regardless of whether their cyclin D1 mRNA expression levels were greater than the median (*p* = 0.288). In the TCGA-COAD (GA) dataset, 28 patients exhibited higher *TCF12* and *MALAT1* gene expression levels but lower *β-catenin* gene expression (i.e., the TCF12^hi^MALAT1^hi^β-catenin^lo^ expression status). Among these patients, those with higher *cyclin D1* gene expression levels had a noticeably poorer OS, although the difference did not reach statistical significance due to the small sample size (*p* = 0.054, Figure 4B). We further defined the patients with a TCF12^hi^MALAT1^hi^β-catenin^lo^cyclin D1^hi^ expression status as having the TMBC pattern. Among the 193 patients, 16 patients exhibited the TMBC pattern, and they had significantly worse OS compared to those without the TMBC pattern (*p* < 0.001, Figure 4C). Since tumor metastasis is a key factor leading to mortality in CRC patients, we validated the association between TMBC pattern and poorer OS in non-metastatic and metastatic patients, respectively. The results revealed that the patients with the TMBC pattern exhibited worse OS outcomes regardless of their metastasis status (Appendix A). Therefore, we further performed univariate Cox regression analyses to assess the impact of both the TMBC pattern and metastasis on OS. The results showed that both the TMBC pattern and metastasis were statistically significant risk factors for poorer OS in CRC patients (Figure 4D). Furthermore, a multivariate Cox regression analysis demonstrated that the TMBC pattern and metastasis were both statistically significant independent risk factors for poorer OS (Figure 4D). The hazard ratio (HR) of the TMBC pattern was even higher than that of metastasis (4.476 vs. 3.517). The nomogram shows that when the TMBC pattern had a prognostic score of 100, tumor metastasis exhibited a relative score of 84 (Figure 4E), suggesting that the TMBC pattern is a metastasis-independent and even stronger prognostic factor for poorer OS in CRC patients. Patients with the TMBC pattern could be predicted to have a one-year survival rate of 60%, but their three-year survival rate dropped to only 30%. These TMBC patients exhibited a worse prognosis regardless of metastasis. We wondered if tumor recurrence could be a primary risk factor contributing to the mortality of this group of patients. As shown in Figure 4F, these TMBC patients indeed had a significantly shorter RFS, suggesting that early tumor recurrence could be a mortality risk for these patients.

## 4. Discussion

Despite advancements in medical treatments that have significantly improved the survival time and quality of life for CRC patients, tumor metastasis remains a critical factor that greatly worsens prognosis. The occurrence of metastasis often marks a pivotal turning point in CRC progression, typically associated with poorer survival outcomes. Currently, the five-year survival rate is approximately 90% for the patients with localized CRC (where cancer has not spread out of the colon or rectum). However, once CRC metastasizes to distant organs, patients’ survival rates drop sharply down to around 14%. Analysis of the available data of 447 patients included in the TCGA-COAD dataset also shows that over 70% of non-metastatic CRC patients survive beyond 5 years, but for those patients with CRC metastasis, their five-year overall survival rate reduces dramatically to less than 30%. The occurrence of metastasis signals a warning for disease progression and the patient’s survival, drawing the attention of most CRC researchers worldwide. However, for those CRC patients who do not develop metastasis but still have poor survival outcomes, the causes and mechanisms of mortality are rarely studied. In our study, we discovered that 8.3% of CRC patients, regardless of whether they experience tumor metastasis, have poor prognosis due to their tumor tissues expressing high TCF12 mRNA, high MALAT1, low β-catenin mRNA, and high cyclin D1 mRNA.

TCF12 is a transcription factor that regulates the RNA expression levels of its target genes. MALAT1 also regulates RNA expression through its roles as a molecular scaffold, guide RNA, and miRNA sponge [17,18]. Our present study has uncovered a TCF12–MALAT1 alliance, as MALAT1 was found to physically associate with TCF12 in CRC cells, and its gene expression was coordinated with that of *TCF12*. Importantly, the expressions of TCF12 mRNA and MALAT1 under this alliance, rather than their individual expressions, were significantly associated with the poorer OS in CRC patients. Although the overexpression of MALAT1 [32] or TCF12 [5] has been independently linked to poorer prognosis in CRC patients, investigations conducted across different cancer types or patient cohorts may have contradictory results. The inconsistencies may partly arise from the involvement of TCF12 and MALAT1 in extensive biological processes mediated by interactions with various regulatory factors. As aforementioned, TCF12 is an E-box-binding transcription factor, with ChIP and DNA sequencing analyses identifying 15,373 candidate downstream target genes [3], indicating its broad regulatory influence. Due to its HLH protein structure, TCF12 can interact with numerous transcription factors, collectively shaping its role in regulating target gene expressions. TCF12 can also interact with lncRNAs that modulate its activity. For example, the lncRNA MIAT binds to TCF12 and facilitates its binding to the promoter region of *NFAT5* gene, thereby activating NFAT5 expression and promoting melanoma cell proliferation, migration, and invasion [13]. In this study, we observed that TCF12 also binds to lncRNA MALAT1, one of the earliest identified and widely studied lncRNAs. According to the LncRNA2Target database, MALAT1 has approximately 2650 predicted target genes. Furthermore, considering MALAT1′s function as a molecular scaffold or guiding RNA to interact with DNA and transcription regulatory proteins [23,24,25,26], the total number of its target genes could far exceed this prediction. Notably, the TCF12–MALAT1 alliance was validated experimentally. When we ectopically overexpressed TCF12 in SW480 cells, MALAT1 expression increased correspondingly, and MALAT1 was found to bind to TCF12 (Appendix A). SW480, a cell line established from a primary CRC tumor, exhibits low TCF12 expression, whereas SW620, a paired cell line established from lymph node metastasis of the same patient, expresses high levels of TCF12 [5]. These experimental results are consistent with our description of the TCF12–MALAT1 alliance. Additionally, MALAT1 contains RNA sequences predicted to form triplex structures with several gene promoter regions, suggesting that MALAT1 may function as a guide RNA or molecular scaffold, facilitating the recruitment of TCF12 to promoter regions of target genes and its interactions with other transcriptional regulatory proteins. Consequently, the TCF12–MALAT1 alliance may produce complex and distinct outcomes that are entirely different from those resulting from the individual activities of TCF12 and MALAT1.

We further explored which downstream factors/events of the TCF12–MALAT1 alliance were responsible for the significant association with patients’ poorer OS outcomes. We performed MGSA signal pathway enrichment analysis on 2312 common target genes of TCF12 and MALAT1 and found that these genes are involved in pathways such as Aurora B, ATM, PLK1, and non-canonical WNT pathways. Given that the WNT–β-catenin pathway is a marker for identifying CMS-2 subtype CRC patients who generally have a relatively better prognosis among all CRC patients [39], the induction of non-canonical WNT pathways (i.e., β-catenin-independent WNT pathways) by the TCF12–MALAT1 alliance may be related to the poorer prognosis of patients. We analyzed the data from 193 patients in the TCGA-COAD (GA) dataset and found that the TCF12 mRNA level was significantly positively correlated with the levels of MALAT1, β-catenin mRNA, and cyclin D1 mRNA, but not with that of c-Myc mRNA. This lack of relationship between the TCF12 and c-Myc mRNA levels was expected, as both belong to the HLH protein family, which involves complex member-to-member interactions [1,2]. However, an interesting and unexpected finding was that the level of MALAT1 was only correlated with the level of cyclin D1 mRNA, not with β-catenin or c-Myc, and the level of β-catenin mRNA was not correlated with the downstream targets of cyclin D1 or c-Myc mRNA expressions. These results suggest that the high expression of cyclin D1 mRNA associated with the TCF12–MALAT1 alliance may occur through a β-catenin-independent pathway. Additionally, an analysis of patients’ OS rates indicated that patients with high expressions of TCF12 mRNA, MALAT1, and β-catenin mRNA had good prognoses, while those with high expressions of TCF12 mRNA, MALAT1, and cyclin D1 mRNA had poor prognoses. Finally, we found that patients with high expressions of TCF12 mRNA, MALAT1, and cyclin D1 mRNA but low expression of β-catenin mRNA—whom we refer to as having the TMBC pattern—had shorter RFS and OS outcomes. Tumor recurrence, rather than metastasis, may be the primary risk factor contributing to the mortality of this group of patients.

The WNT signaling pathways, essentially involved in numerous biological and pathological processes, are commonly classified into the canonical WNT–β-catenin pathway and the non-canonical β-catenin-independent WNT pathways [47]. The canonical WNT pathway primarily involves the binding of WNT ligands with Frizzled (FZD) receptors and LRP5/6 co-receptors, which prevents β-catenin from degradation by the APC–Axin–GSK-3β complex. The stabilized β-catenin then translocates into the nucleus and partners with other transcription factors (e.g., TCF/LEF) to turn on downstream gene expressions for cell proliferation, survival, and stem cell maintenance. The non-canonical WNT pathways exhibit signaling cascades without the participation of β-catenin and as yet include two branches: the WNT/planar cell polarity (PCP) pathway and the WNT/Ca^2+^ pathway [47]. In the WNT/PCP pathway, WNT ligands bind to FZD receptors along with ROR1/2 or RYK co-receptors, activating downstream molecules RhoA and Rac1, which further regulate cell polarity, migration, and tissue morphogenesis. The WNT/Ca^2+^ pathway involves the binding of WNT to FZD, leading to phospholipase C activation and subsequent Ca^2+^ release. Increased Ca^2+^ levels activate Ca^2+^-dependent proteins, such as phosphatase calcineurin, calmodulin-dependent protein kinase II, and protein kinase C, which play roles in cell adhesion, migration, and gene regulation. In our present study, the TCF12–MALAT1 alliance is significantly associated with a β-catenin-independent induction of *cyclin D1* gene expression, which acts as a risk factor for CRC patient mortality, associated more with tumor recurrence than with metastasis. It needs to be clarified whether there are more interactions between the TCF12–MALAT1 alliance and the WNT pathways. With further research, the findings may potentially represent a new branch of the non-canonical WNT pathways.

Cancer is a complicated disease with diverse causes. Even tumors originating from the same organ can exhibit heterogeneity across different patients due to variations in genetic, epigenetic, and tissue microenvironmental factors. This underlines the need for the development of personalized or precision medicine. In our study, we identified two distinct subgroups within CRC patients. One group, comprising 17.1% of patients, showed high levels of TCF12 mRNA, MALAT1, and β-catenin mRNA in tumors, similar to the CMS-2 subtype, and displayed better overall survival; we termed this the “Pan-CMS-2 pattern” (Figure 5). Another group, encompassing 8.3% of patients, exhibited a TMBC pattern with high expressions of TCF12 mRNA, MALAT1, and cyclin D1 mRNA, but low β-catenin mRNA, and had shorter RFS and OS outcomes regardless of whether metastasis occurs. Tumor recurrence seemed to be a major mortality risk for these CRC patients with the TMBC pattern. Cyclin D1 is a key cell cycle regulatory protein that can form complexes with CDK4 and CDK6, thereby activating their kinase activities and promoting G1 to S phase progression in the cell cycle [48]. Several drugs targeting cyclin D1-associated CDK4/6 have been developed and used in clinics. Examples include palbociclib (Ibrance; Pfizer Inc., New York, NY, USA), abemaciclib (LY2835219; Eli Lilly & Co., Indianapolis, IN, USA), and ribociclib (LEE011; Novartis International AG, Basel, Switzerland), which are highly specific oral inhibitors of CDK4/6 [49]. Our study suggests that administering CDK4/6 inhibitors to CRC patients with the TMBC pattern may potentially reduce the risk of early mortality.

There are two limitations of this study. First, the findings were derived from only one dataset, and additional CRC patient datasets are needed to validate the clinical implications of the TMBC pattern. Second, this study lacks investigations into the exact mechanism underlying the simultaneous suppression of *β-catenin* gene expression and upregulation of *cyclin D1* gene expression under the TCF12–MALAT1 alliance. Understanding this mechanism could pave the way for developing preventive strategies. Nonetheless, these limitations do not diminish the potential of the TMBC pattern to serve as a biomarker for evaluating the efficacy of precision therapies using cyclin D1-associated CDK4/6 inhibitors.

## 5. Conclusions

Metastasis is a critical event that significantly worsens the prognosis of CRC patients and has garnered the attention of CRC researchers worldwide. However, for CRC patients who do not develop metastasis but still experience poor survival outcomes, the underlying causes and mechanisms of mortality remain poorly understood. In this study, we identified a subset (8.3%) of CRC patients who, irrespective of tumor metastasis, have a poorer prognosis due to their tumor tissues exhibiting high TCF12 mRNA, high MALAT1, low β-catenin mRNA, and high cyclin D1 mRNA levels, referred to as the TMBC pattern. For these patients, early tumor recurrence, rather than metastasis, appears to be the primary risk factor for mortality. Cyclin D1 is an important cell cycle regulatory protein responsible for activating CDK4 and CDK6 kinase activities and promoting cancer cell proliferation. A precision medicine administering CDK4/6 inhibitors may potentially reduce the risk of early mortality in CRC patients with the TMBC pattern.

## Figures and Tables

**Figure 1 cells-13-02035-f001:**
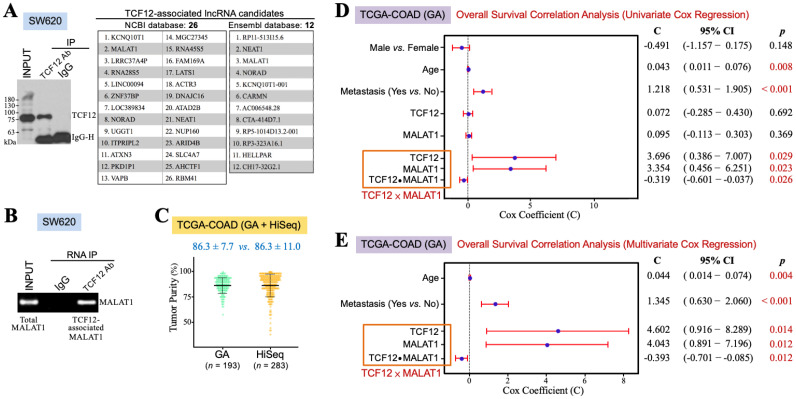
TCF12 cooperates with MALAT1 to exacerbate CRC prognosis. (**A**) Identification of MALAT1 as a TCF12-associated lncRNA by RNA immunoprecipitation (IP) and sequencing. After immunoblot analysis to confirm that TCF12 was specifically immunoprecipitated by the anti-TCF12 antibody from the lysate of SW620 cells, the anti-TCF12 immunoprecipitates were further subjected to the procedures of RNA extraction and next-generation sequencing. Several TCF12-associated lncRNAs were annotated after the sequence alignment analyses with NCBI and Ensembl databases. (**B**) Validation of MALAT1 as a TCF12-associated lncRNA. MALAT1 was detected by RT-PCR from the RNA sample isolated from the anti-TCF12 immunoprecipitates of SW620 cells. (**C**) ESTIMATE algorithm was used to estimate the tumor purities of the CRC specimens employed by the GA and HiSeq platforms of the TCGA-COAD dataset. (**D**) Univariate Cox regression analyses showing that age, metastasis, and expression levels of TCF12 mRNA and MALAT1 under an alliance but not alone were significantly associated with patients’ shorter overall survival outcomes. The interaction term of TCF12 mRNA and MALAT1 expression levels, designated as “TCF12 × MALAT1”, was represented as three covariates in a multivariate Cox regression analysis: TCF12 mRNA expression level (designated as “TCF12”), MALAT1 expression level (designated as “MALAT1”), and the multiplication of TCF12 mRNA and MALAT1 expression levels (designated as “TCF12 ⦁ MALAT1”). (**E**) Multivariate Cox regression analysis showing that the association of the TCF12–MALAT1 alliance with CRC poorer prognosis remained statistically significant after adjusting for patients’ age and tumor metastasis.

**Figure 2 cells-13-02035-f002:**
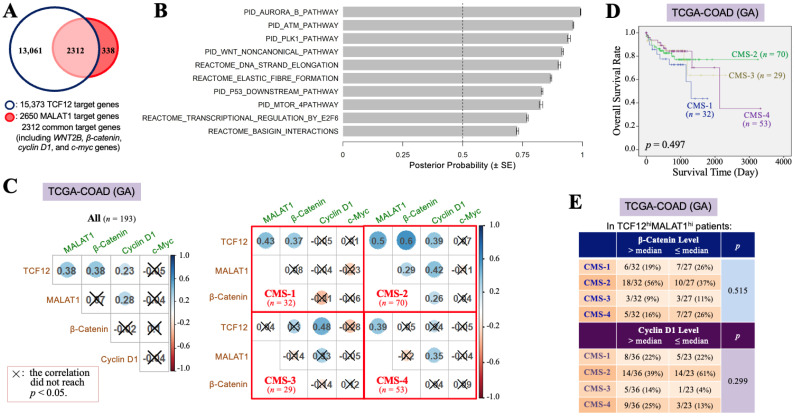
The TCF12–MALAT1 alliance is associated with *β-catenin*-independent *cyclin D1* expression. (**A**) A group of 2312 genes, designated as TCF12 and MALAT1 common target genes, were obtained by examining the overlap between the total 15,373 genes identified from anti-TCF12 ChIP and DNA sequencing [3] and the list of 2650 MALAT1 target genes of the UALCAN website (University of Alabama at Birmingham) [45]. They include *WNT2B*, *β-catenin*, *cyclin D1*, and *c-myc* genes. (**B**) The 2312 genes were analyzed by MGSA signal pathway enrichment. The significantly enriched pathways contain the non-canonical WNT pathway. (**C**) Pearson’s correlation analyses of the levels of TCF12 mRNA, MALAT1, β-catenin mRNA, cyclin D1 mRNA, and c-Myc mRNA in all 193 patients or the different subsets of patients classified based on the CMS criteria. (**D**) Kaplan–Meier OS curves of the 184 TCGA-COAD (GA) patients classified into 4 subtypes of the CMS system. The CMS-2 patients exhibited a better prognosis, but the CMS-1 and CMS-4 patients had relatively low OS rates. (**E**) The expression statuses of β-catenin and cyclin D1 mRNA in different CMS subtypes of the 59 patients with higher levels (>median) of TCF12 mRNA and MALAT1 expressions (designated as “TCF12^hi^MALAT1^hi^”). The CMS-2 patients trended to express higher levels (>median) of β-catenin mRNA but not cyclin D1 mRNA; however, other subtypes of patients seemed to express higher levels (>median) of cyclin D1 mRNA instead of β-catenin mRNA.

**Figure 3 cells-13-02035-f003:**
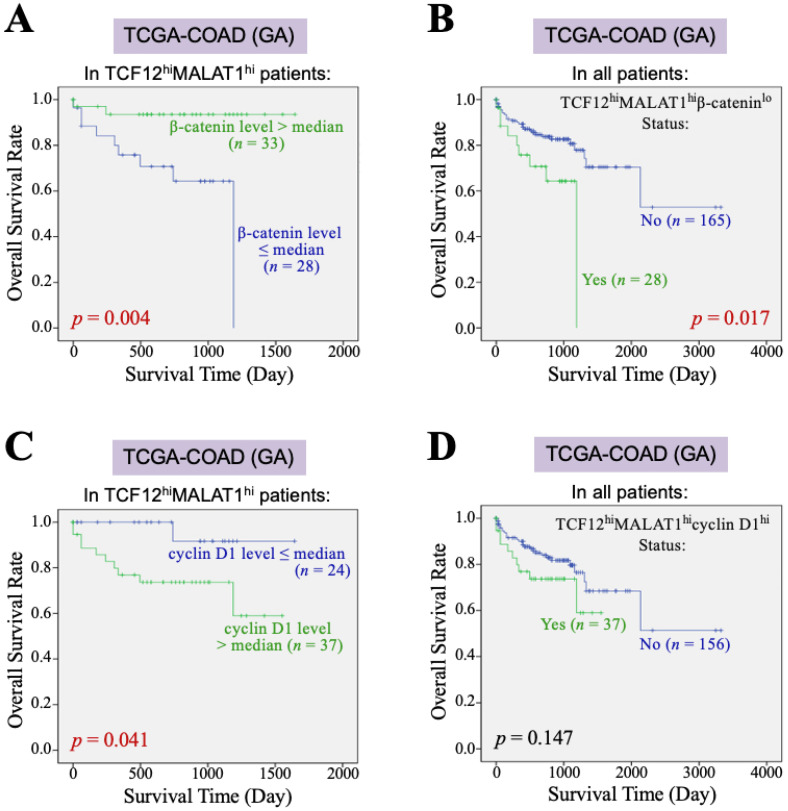
*β-catenin* and *cyclin D1* gene expression affect the association of the TCF12–MALAT1 alliance with CRC patients’ poorer survival outcomes. (**A**) Kaplan–Meier OS curves of the 61 TCGA-COAD (GA) TCF12^hi^MALAT1^hi^ patients who were divided into two groups based on the β-catenin mRNA expression level > or ≤ median. Low β-catenin mRNA expression was significantly associated with shorter OS in the TCF12^hi^MALAT1^hi^ patients (*p* = 0.004 by the log-rank test). (**B**) Kaplan–Meier OS curves of the 193 TCGA-COAD (GA) patients who were divided into two groups based on the patients with or without the status of higher TCF12 mRNA expression, higher MALAT1 expression, but low β-catenin mRNA expression (designated as “TCF12^hi^MALAT1^hi^β-catenin^lo^”). The patients with the TCF12^hi^MALAT1^hi^β-catenin^lo^ expression status significantly exhibited a poorer prognosis when compared with other patients (*p* = 0.017 by the log-rank test). (**C**) Kaplan–Meier OS curves of the 61 TCGA-COAD (GA) TCF12^hi^MALAT1^hi^ patients who were divided into two groups based on the cyclin D1 mRNA expression level > or ≤ median. The patients with high cyclin D1 mRNA expression had a worse OS rate (*p* = 0.041 by the log-rank test). (**D**) Kaplan–Meier OS curves of the 193 TCGA-COAD (GA) patients who were divided into two groups based on the patients with or without the status of higher TCF12 mRNA expression, MALAT1 expression, and cyclin D1 mRNA expression (designated as “TCF12^hi^MALAT1^hi^cyclin D1^hi^”). The patients’ OS between the two groups showed no significant difference.

**Figure 4 cells-13-02035-f004:**
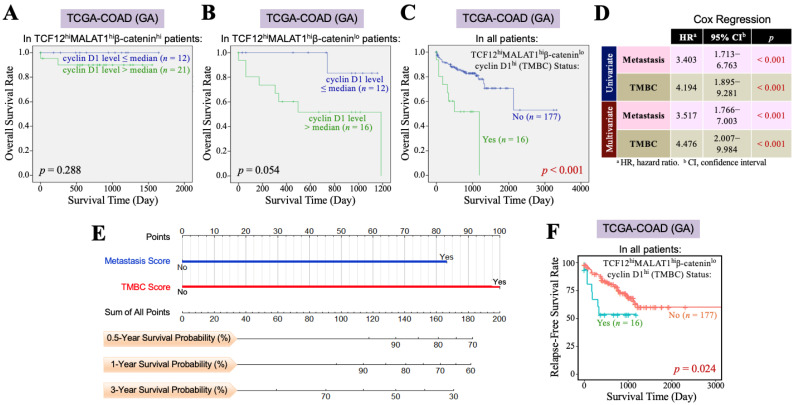
The TCF12–MALAT1 alliance exacerbates CRC prognosis via low *β-catenin* but high *cyclin D1* gene expression. (**A**) Kaplan–Meier OS curves of the 33 TCGA-COAD (GA) TCF12^hi^MALAT1^hi^β-catenin^hi^ patients who were divided into two groups based on the cyclin D1 mRNA expression level> or ≤median. High cyclin D1 mRNA expression did not significantly worsen the OS of TCF12^hi^MALAT1^hi^β-catenin^hi^ patients. (**B**) Kaplan–Meier OS curves of the 28 TCGA-COAD (GA) TCF12^hi^MALAT1^hi^β-catenin^lo^ patients who were divided into 2 groups based on the cyclin D1 mRNA expression level> or ≤median. High cyclin D1 mRNA expression rendered the TCF12^hi^MALAT1^hi^β-catenin^lo^ patients prone to shorter survival. (**C**) Kaplan–Meier OS curves of the 193 TCGA-COAD (GA) patients who were divided into two groups based on the patients with or without the status of higher TCF12 mRNA expression, higher MALAT1 expression, low β-catenin mRNA expression, and higher cyclin D1 mRNA expression (designated as “TCF12^hi^MALAT1^hi^β-catenin^lo^cyclin D1^hi^” or “TMBC pattern”). The patients with the TMBC pattern significantly exhibited a poorer prognosis when compared with other patients (*p* < 0.001 by the log-rank test). (**D**) Univariate and multivariate Cox regression analyses showing that the TMBC pattern is a metastasis-independent prognostic factor with an even higher hazard ratio. (**E**) A nomogram arose based on the effect levels of TMBC vs. metastasis on the OS of 193 TCGA-COAD (GA) patients. When the TMBC score was 100, metastasis reached a score of 83, suggesting that the TMBC pattern serves as a higher risk factor rather than metastasis. (**F**) Kaplan–Meier RFS curves of the 193 TCGA-COAD (GA) patients who were divided into two groups based on the patients with or without the TMBC pattern. The patients with the TMBC pattern significantly exhibited a shorter RFS outcome when compared with other patients (*p* = 0.024 by the log-rank test).

**Figure 5 cells-13-02035-f005:**
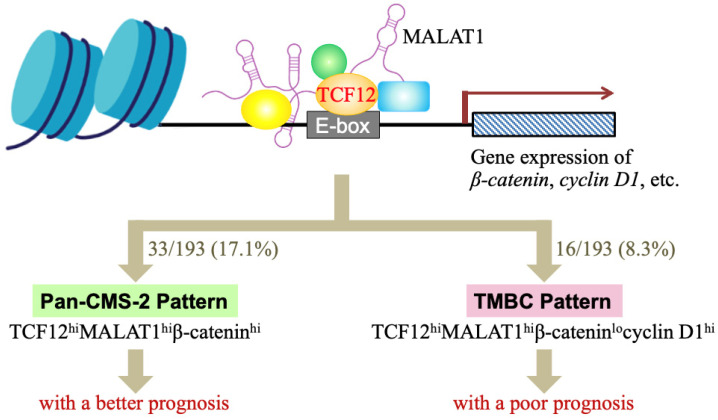
A schematic illustration summarizing our studies of whether the TCF12–MALAT1 alliance and its downstream β-catenin and cyclin D1 determine a good or poor CRC prognosis. TCF12 is a transcriptional factor involved in numerous cellular processes. MALAT1 is a long lncRNA that can form a complex with TCF12, creating an alliance with correlated expression levels in CRC patients. This TCF12–MALAT1 alliance is linked to poorer prognosis independently of age and metastasis status. To identify the downstream factors/events responsible for this outcome, we analyzed 2312 common target genes of TCF12 and MALAT1 through MGSA pathway enrichment analysis, finding involvement in pathways like Aurora B, ATM, PLK1, and non-canonical WNT. We further investigated the impact of WNT downstream genes *β-catenin* and *cyclin D1* on survival in CRC patients with the TCF12–MALAT1 alliance. Our analysis of the TCGA-COAD (GA) dataset revealed that 61 out of 193 patients had tumors with the “TCF12^hi^MALAT1^hi^” expression status. Furthermore, 33 out of 61 patients had higher β-catenin mRNA levels in tumors. They were classified as having a “Pan-CMS-2 pattern” and exhibited better survival outcomes. For the other 28 patients with low expression levels of β-catenin mRNA in tumors, 16 out of them expressed higher levels of cyclin D1 mRNA in tumors, i.e., the “TCF12^hi^MALAT1^hi^β-catenin^lo^cyclin D1^hi^” expression status or the so-called “TMBC pattern”. They exhibited poorer RFS and OS outcomes independently of metastasis. Early tumor recurrence seemed to be a mortality risk to these TMBC patients.

**Table 1 cells-13-02035-t001:** Elevation of TCF12 mRNA expression is significantly correlated with increased expression of MALAT1 in the CRC specimens of TCGA-COAD (GA) dataset.

	TCF12^hi^ Status ^#^		*p*-Value
	No (*n* = 97)	Yes (*n* = 96)	
Gender (male/female)	50/47	44/52	0.427
Age (mean ± SD, year)	67.82 ± 12.76	71.54 ± 11.81	**0.037**
TNM staging (I/II/III/IV)	16/37/28/16	20/39/23/14	0.773
Tumor recurrence (no/yes)	73/24	70/26	0.711
Metastatic occurrence (no/yes)	53/44	59/37	0.337
MALAT1^hi^ status ^##^ (no/yes)	62/35	35/61	**<0.001**
TCF12 mRNA level (mean ± SD)	9.57 ± 0.62	10.79 ± 0.52	**<0.001**
MALAT1 level (mean ± SD)	11.49 ± 1.43	12.31 ± 1.34	**<0.001**

**^#^** The TCF12^hi^ Status is defined for those patients with CRC tissues expressing TCF12 mRNA levels > median. ^##^ The MALAT1^hi^ Status is defined for those patients with CRC tissues expressing MALAT1 levels > median.

**Table 2 cells-13-02035-t002:** Elevation of both TCF12 mRNA and MALAT1 expressions is significantly associated with increased expression of cyclin D1 mRNA in the CRC specimens of TCGA-COAD (GA) dataset.

	TCF12^hi^MALAT1^hi^ Status ^#^		*p*-Value
	No (*n* = 132)	Yes (*n* = 61)	
Gender (male/female)	65/67	29/32	0.826
Age (mean ± SD, year)	68.77 ± 12.77	71.62 ± 11.44	0.124
TNM staging (I/II/III/IV)	22/55/34/21	14/21/17/9	0.672
Tumor recurrence (no/yes)	97/35	46/15	0.777
Metastatic occurrence (no/yes)	77/55	35/26	0.900
TCF12 mRNA level (mean ± SD)	9.90 ± 0.82	10.78 ± 0.50	**<0.001**
MALAT1 level (mean ± SD)	11.36 ± 1.33	13.07 ± 0.86	**<0.001**
β-Catenin mRNA level (mean ± SD)	13.56 ± 0.56	13.69 ± 0.45	0.108
Cyclin D1 mRNA level (mean ± SD)	11.93 ± 0.60	12.27 ± 0.68	**0.001**

**^#^** The TCF12^hi^MALAT1^hi^ Status is defined for those patients with CRC tissues expressing both TCF12 mRNA and MALAT1 levels more than their respective median values.

## Data Availability

The RNA sequencing data of anti-TCF12 immunoprecipitates are deposited with an accession No. PRJNA1190159 in the Sequence Read Archive (SRA) database, National Center for Biotechnology Information (NCBI), USA. Other data generated or analyzed during this study are included in this published article and the associated online Appendix A.

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
