# Peer review of "TCF12 and LncRNA MALAT1 Cooperatively Harness High Cyclin D1 but Low β-Catenin Gene Expression to Exacerbate Colorectal Cancer Prognosis Independently of Metastasis"

_cells, 2024, doi:10.3390/cells13242035_

Round 1
Reviewer 1 Report
Comments and Suggestions for Authors
This manuscript evaluated the prognostic significance of increased expression of MALAT1 and TCF12, as well as other genes, in colorectal cancer. In vitro immunoprecipitation indicated physical association of MALAT1 and TCF12. Genomic express data from a set of 193 patients indicated that patients with nominally increased expression of both MALAT1 and TCF12 had either more favorable prognosis when beta-catenin expression was increased, or poor prognosis when beta-catenin expression was decreased if cyclin D1 expression was also increased relative to mean expression levels. These results suggest a non-canonical WNT pathway process is mediated by increased expression of TCF12 and MALAT1. Overall, the data is presented clearly in the text and in the figures, and conclusions align with the data as shown. Review of the manuscript raised the following points and comments for the authors to consider.
Points to be addressed:
-Please explain why TCGA-COAD data (from a 2016 dataset) were used for gene expression analyses, rather than all available colorectal cancer tumor data in the updated TCGA database?
-Have the authors considered using the proposed TMBC pattern of gene expression in an additional data set as a validation of this potential prognostic indicator? Considering that multiple comparisons were done between sub-cohorts of patients (Figures 3 and 4) using only a single data set with relatively limited number of patients, it could be helpful to validate the prognostic significance of the TMBC pattern using an additional data set.
-In Figure 1D and 1E, please clarify what is indicated by “TCF12 x MALAT1”.
Author Response
Comment-1: Please explain why TCGA-COAD data (from a 2016 dataset) were used for gene expression analyses, rather than all available colorectal cancer tumor data in the updated TCGA database?
[RESPONSE]: We used the RNA sequencing and patients’ clinical data included in the UCSC Xena platform (Goldman et al. Nat. Biotechnol. 2020, 38, 675-678) in which RNA sequencing data were a version issued on 10/13/2017 and patients’ survival data were curated by Liu et al. in 2018 (Liu et al. Cell 2018, 173, 400-416.e11). We have made extra statements in the 2.2. section (lines 119-134) of the revised manuscript.
Comment-2: Have the authors considered using the proposed TMBC pattern of gene expression in an additional data set as a validation of this potential prognostic indicator? Considering that multiple comparisons were done between sub-cohorts of patients (Figures 3 and 4) using only a single data set with relatively limited number of patients, it could be helpful to validate the prognostic significance of the TMBC pattern using an additional data set.
[RESPONSE]: We sincerely thank Reviewer #1 for this important comment. We have made extensive efforts to identify an additional CRC patient dataset to validate the clinical implication of the TMBC pattern described in our study. As part of these efforts, we analyzed microarray data from the KM Plotter database. Our analysis revealed that higher tumor MALAT1 expression is significantly associated with shorter RFS and OS in CRC patients (the results shown below). However, higher tumor expressions of TCF12, β-catenin, and cyclin D1 mRNA are not associated with OS but are significantly associated with better RFS in CRC patients. These results suggest that an interaction with MALAT1 expression may alter the prognostic impact of elevated TCF12 gene expression in CRC patients. Unfortunately, the KM Plotter database does not provide access to raw data or tools for multiple Cox regression analysis, preventing us from further investigating the clinical implications of the TMBC or Pan-CMS-2 pattern in CRC patients. Additionally, our analyses of the Oncomine database demonstrated a significant positive correlation between tumor TCF12 mRNA and MALAT1 expression levels in the Kaiser colon dataset. However, this dataset lacks survival data, limiting further prognostic analyses. While Reviewer #2 suggested using the OSdream database, we found that it only includes data from other cancers, such as breast invasive carcinoma, leiomyosarcoma, myxofibrosarcoma, skin cutaneous melanoma, and uveal melanoma, and does not contain CRC patient data. As a result, we have not been able to use additional datasets to validate the clinical implications of the TMBC pattern at this time. To address this, we have added a new paragraph on the bottom of the Discussion section to acknowledge the limitations in our current study (lines 576-584). Thank you for your understanding.

Comment-3: In Figure 1D and 1E, please clarify what is indicated by “TCF12 x MALAT1”.
[RESPONSE]: To comprehensively evaluate the contribution of the TCF12-MALAT1 alliance to CRC prognosis, we conducted Cox regression analyses on the interaction term of TCF12 mRNA and MALAT1 expressions, referred to as “TCF12´MALAT1”. We have added extra statements in the 2.6. section (lines 198-204), 3.1. section (lines 239-244), and Figure 1D legend (lines 268-273) of the revised manuscript to clearly indicate the “TCF12 x MALAT1”.

Reviewer 2 Report
Comments and Suggestions for Authors
Dear Authors, an interesting study on the TCF12-MALAT1 alliance in CRC. However, substantial amendments are required before proceeding further. Thus, please answer or consider the following:
(1) Abstract, lines 27-28: consider changing “β-catenin and cyclin D1” to actual gene symbols when referring to them on the DNA level. This would probably change “TMBC” to “TMCC” throughout the paper because β-catenin is encoded by CTNNB1 while cyclin D1 by CCND1. If you want to refer to these genes later as β-catenin and cyclin D1, add an explanation on the first use of symbols.
(2) Introduction, line 40: are you sure that TCF12 regulates 15,373 genes? Are these genetic targets confirmed experimentally, all via ChIP? This is quite a huge number of targets, even for a transcription factor.
(3) Introduction, line 50: Sometimes I think that there are double space marks between sentences. Please double-check the entire paper.
(4) Introduction, lines 94-117: from my understanding, this is the recapitulation of the present study (I did not find any citation and you summarized your own observations). If yes, I think it should be condensed a bit because, in the present form, it fits better in Discussion rather than Introduction. The entire Introduction could be gravitated towards the background of your research scope. Improvements in the last paragraphs are mandatory.
(5) Materials and Methods, lines 125-128: section 2.2 would benefit from more details. Mention what kind of TCGA data were acquired from Xena (e.g., which protocol – STAR or HTSeq; moreover – were data raw or processed; what kind of normalization, etc.). If you investigated the clinical implications, you need to rephrase the sentence where you mention that RNA sequencing data were acquired (in fact, they were acquired alongside clinical data – state which one).
(6) Materials and Methods, line 173: why hg19 was used as a reference instead of GRCh38/hg38 or even T2T-CHM13v2.0? It is quite outdated. Were there technical obstacles?
(7) Materials and Methods, lines 159-177: were data from the experiment described in section 2.4 uploaded to the relevant repository? In my opinion, it is obligatory to deposit these data before the manuscript can proceed. Afterwards, change the “Data Availability Statement” (lines 566-567) accordingly.
(8) Materials and Methods, line 177: was fold-change the only requirement? Did you require a specific p-value or FDR?
(9) Materials and Methods, line 189: why did you choose OS as the only endpoint? Endpoints such as DFS/RFS/PFS offer earlier presentation since events due to disease recurrence/progression occur earlier than death from the disease. This is especially important if you investigated metastasis; it would be beneficial in parts of the study such as in Figure 4D (lines 366-368). It would be ideal to use MFS (metastasis-free survival) but since you incorporated TCGA data, it might be hard to do so (from my expertise, the so-called OSdream database includes MFS only for specific GEO datasets).
(10) Results: your descriptions fit better in the case of the “Results and Discussion” section than “Results”. There are many inclusions that do not explain results, but rather serve as a background. Either limit such inclusions or consider changing the section to “Results and Discussion”. In addition, contact a native speaker when improving your manuscript. There are also descriptions that should be put in the methodology of your paper, e.g., the one about batch effects (lines 213-216), where ESTIMATE is mentioned for the first time whereas I think it should be in methods.
(11) Results, line 207: the use of a single cell line (SW620) is unusual and should be considered as a limitation of your study. Please prepare a separate paragraph recapitulating the strengths and limitations of your work; preferably somewhere at the end of the main text.
(12) Results, lines 213-216: I thought that RUVSeq is used to address batch effects and not ESTIMATE, which is related to the tumor microenvironment?
(13) Results, lines 219-221: From my understanding, a specific sample has GA data or HiSeq data in the TCGA-COAD cohort (see https://www.biostars.org/p/174112/#174211). So why focus on GA only (even if ESTIMATE scores had smaller variation), if you lose a lot of data that could be helpful in subsequent steps such as calculations in Figure 2E?
(14) Results, lines 227-228: the explanation of “TCF12-MALAT1 alliance” could be mentioned earlier in the text, or at least repeated.
(15) Results, line 236: why there are “TCF12” and “MALAT1” twice in Figure 1D? This is crucial since their effect seems to be different. In Figure 1E, “TCF12” and “MALAT1” are shown once, similar to their alliance.
(16) Results, lines 251-253: contact the journal’s Office if significant p-values could be marked with red font or maybe it would be better to make it bold and with normal black font. Alternatively, there could be a normal black font and significance marks next to the value. The same could be done in Figures.
(17) Results, lines 260-261: Figure 2A should be visualized as a Venn diagram or upset-plot. Currently, it is less scientific. Moreover, from my understanding – TCF12 targets are on the DNA level whereas those of MALAT1 are on the RNA level? This must be described in the text. Did take into account the possibility of creating an RNA-DNA hybrid in the case of MALAT1? This would allow you to indicate an even more important set of genes which are regulated by MALAT1 on both DNA and RNA levels.
(18) Results, lines 263-264: are all pathways significant in Figure 2B? Is 0.5 value a threshold of significance?
(19) Results, Figure 2C: are colors necessary? I think it would be much more valuable to color only significant results correlations and use a background color of either red (for positive correlation) or blue (for negative correlation). This would be much more self-explanatory while not overwhelming.
(20) Results, lines 278-282: information about acquiring CMS classification should be mentioned in the methodology. Ideally, you should explain all subtypes in Methods whereas in Results there should be only a percentage of each subtype without its molecular description.
(21) Results, line 295: pairwise p-values would be much more valuable, because probably CMS1 vs CMS2 or CMS4 comparison is significant.
(22) Results, lines 341-342: how were tumor recurrence and metastasis included if the only endpoint was overall survival? Or maybe they were assessed on the basis of other data than survival?
(23) Results, line 361: you can mention that “TMBC” (or “TMCC”, if you address my first comment) comes from the first letter or each gene symbol.
(24) Discussion, line 442: delete “long” before “lncRNA”, because the meaning is already in the abbreviation.
(25) Discussion, lines 483-484: is this possible to suggest what could be the primary risk factor if not metastasis? Given the biological functions of gene products of TCF12/MALAT1/b-catenin/cyclinD1?
(26) Discussion, lines 485-513: I think that the description of Wnt signaling is too long and deviates from the topic. Please reduce it a bit.
(27) The “Conclusion” section is missing while I think the manuscript would benefit from the simple take-home message from your study.
Comments on the Quality of English LanguageModerate improvements are required.
Author Response
Comment-1: Abstract, lines 27-28: consider changing “β-catenin and cyclin D1” to actual gene symbols when referring to them on the DNA level. This would probably change “TMBC” to “TMCC” throughout the paper because β-catenin is encoded by CTNNB1 while cyclin D1 by CCND1. If you want to refer to these genes later as β-catenin and cyclin D1, add an explanation on the first use of symbols.
[RESPONSE]: Thank you for pointing this out. We have updated the manuscript by including the gene names CTNNB1 and CCND1 when β-catenin and cyclin D1 are first mentioned (lines 27–28 in the Abstract section and the line 296 in the 3.2. section). This ensures clarity and consistency for readers, particularly those who may not be familiar with the corresponding gene symbols. Thank you for your suggestion!
Comment-2: Introduction, line 40: are you sure that TCF12 regulates 15,373 genes? Are these genetic targets confirmed experimentally, all via ChIP? This is quite a huge number of targets, even for a transcription factor.
[RESPONSE]: The 15,373 so-called TCF12 target genes were adopted from the ENCODE Transcription Factor Targets Dataset, based on ChIP and DNA sequencing data. It is important to note that not all of these genes have been individually validated through experimental studies. However, in our laboratory, we have done experiments on several genes, including MALAT1, CTNNB1, CDH1, CDH5, and COL1A1, and confirmed the existence of TCF12-binding sites in their promoter regions. Additionally, a new reference #16 indicates a recent study published in Pancreatology in which validated RMND5A as a TCF12 target gene (Zhou et al. Pancreatology 2024, 24, 1073-1083)(line 64 in the Introduction section).
Comment-3: Introduction, line 50: Sometimes I think that there are double space marks between sentences. Please double-check the entire paper.
[RESPONSE]: That's correct; I double-spaced between sentences as part of the practice I learned during my postdoctoral training in the U.S. I fully respect the Editorial Office's formatting adjustments for the manuscript.
Comment-4: Introduction, lines 94-117: from my understanding, this is the recapitulation of the present study (I did not find any citation and you summarized your own observations). If yes, I think it should be condensed a bit because, in the present form, it fits better in Discussion rather than Introduction. The entire Introduction could be gravitated towards the background of your research scope. Improvements in the last paragraphs are mandatory.
[RESPONSE]: Thanks for this comment. We have condensed the last paragraph in the revised manuscript.
Comment-5: Materials and Methods, lines 125-128: section 2.2 would benefit from more details. Mention what kind of TCGA data were acquired from Xena (e.g., which protocol – STAR or HTSeq; moreover – were data raw or processed; what kind of normalization, etc.). If you investigated the clinical implications, you need to rephrase the sentence where you mention that RNA sequencing data were acquired (in fact, they were acquired alongside clinical data – state which one).
[RESPONSE]: We adopted the RNA sequencing and patients’ clinical data from the UCSC Xena platform (Goldman et al. Nat. Biotechnol. 2020, 38, 675-678) in which RNA sequencing data were a version issued on 10/13/2017 and patients’ survival data were curated by Liu et al. in 2018 (Liu et al. Cell 2018, 173, 400-416.e11). The RNA sequencing data, transformed into log2(x+1) RSEM normalized counts, were downloaded and used for subsequent analyses without any preprocessing. The clinical information including patient’s gender, age, pathologic stage was downloaded from the Phenotypes file, and the relapse-free survival (RFS) and overall survival (OS) were determined based on the curated survival data available through Xena. We have added extra statements in the revised 2.2. section (lines 119-134).
Comment-6: Materials and Methods, line 173: why hg19 was used as a reference instead of GRCh38/hg38 or even T2T-CHM13v2.0? It is quite outdated. Were there technical obstacles?
[RESPONSE]: To identify TCF12-associated lncRNAs, we performed RNA profiling on IgG-IP and TCF12-IP samples. After RNA sequencing, the reads were mapped to the hg19 reference genome. This analysis had been done in 2016-2017. We fully acknowledge that newer genome versions (GRCh38/hg38 or even T2T-CHM13v2.0) provide more comprehensive genomic annotations. While the mapping efficiency using hg19 may be lower than with newer references, this should not significantly impact our current results. Moreover, we explore their expression levels using the TCGA database, which predominantly relies on hg19 as the reference genome. Using newer genome versions might allow for the identification of additional lncRNA candidates, but these candidates may not be represented in the TCGA database. Considering this, we analyzed the TCF12-IP data using hg19 as the reference genome.
Comment-7: Materials and Methods, lines 159-177: were data from the experiment described in section 2.4 uploaded to the relevant repository? In my opinion, it is obligatory to deposit these data before the manuscript can proceed. Afterwards, change the “Data Availability Statement” (lines 566-567) accordingly.
[RESPONSE]: We have uploaded the raw data to the SRA database (Accession No. PRJNA1190159), and also updated “Data Availability Statement”.
Comment-8: Materials and Methods, line 177: was fold-change the only requirement? Did you require a specific p-value or FDR?
[RESPONSE]: p < 0.05 was a threshold of significance.
Comment-9: Materials and Methods, line 189: why did you choose OS as the only endpoint? Endpoints such as DFS/RFS/PFS offer earlier presentation since events due to disease recurrence/progression occur earlier than death from the disease. This is especially important if you investigated metastasis; it would be beneficial in parts of the study such as in Figure 4D (lines 366-368). It would be ideal to use MFS (metastasis-free survival) but since you incorporated TCGA data, it might be hard to do so (from my expertise, the so-called OSdream database includes MFS only for specific GEO datasets).
[RESPONSE]: We also analyzed the association between the TCF12-MALAT1 alliance and RFS, finding a similar trend as with OS. However, the results did not reach statistical significance (lines 252-255 and the extra Supplemental Figure 2). Therefore, we used OS as our primary endpoint. Additionally, we reviewed the OSdream database and found that among the cancer types available for MFS analysis, only breast invasive carcinoma, leiomyosarcoma, myxofibrosarcoma, skin cutaneous melanoma, and uveal melanoma were included. Unfortunately, colorectal cancer was not available for analysis.
Comment-10: Results: your descriptions fit better in the case of the “Results and Discussion” section than “Results”. There are many inclusions that do not explain results, but rather serve as a background. Either limit such inclusions or consider changing the section to “Results and Discussion”. In addition, contact a native speaker when improving your manuscript. There are also descriptions that should be put in the methodology of your paper, e.g., the one about batch effects (lines 213-216), where ESTIMATE is mentioned for the first time whereas I think it should be in methods.
[RESPONSE]: As suggested, we have moved the statements regarding ESTIMATE methodology into 2.2. section (lines 125-130).
Comment-11: Results, line 207: the use of a single cell line (SW620) is unusual and should be considered as a limitation of your study. Please prepare a separate paragraph recapitulating the strengths and limitations of your work; preferably somewhere at the end of the main text.
[RESPONSE]: We did confirm that TCF12 can indeed bind to MALAT1 by using another cell line SW480 with ectopic TCF12 overexpression. We have added the data as Supplemental Figure 1 and made extra statements in the lines 223-225 of the 3.1. section and the lines 494-500 of the Discussion section.

Comment-12: Results, lines 213-216: I thought that RUVSeq is used to address batch effects and not ESTIMATE, which is related to the tumor microenvironment?
[RESPONSE]: We used the ESTIMATE method, based on single-sample GSEA (ssGSEA), to estimate tumor purity. ssGSEA is a rank-based approach that does not require RNA-seq expression levels but instead uses the ranking of a specific gene among 10,412 genes. This rank-based calculation for each sample significantly mitigates batch effects. We were uncertain whether applying RUVSeq to address batch effects prior to purity estimation and subsequent association analysis might introduce unnecessary bias. Therefore, we directly used the log2(x+1) RSEM normalized counts of RNA sequencing data for our calculations.
Comment-13: Results, lines 219-221: From my understanding, a specific sample has GA data or HiSeq data in the TCGA-COAD cohort (see https://www.biostars.org/p/174112/#174211). So why focus on GA only (even if ESTIMATE scores had smaller variation), if you lose a lot of data that could be helpful in subsequent steps such as calculations in Figure 2E?
[RESPONSE]: We did analyze the total data from both GA and HiSeq platforms. The data are shown below. We did not see any significant improvement.
|
|
Beta-Catenin > median |
Beta-Catenin <= median |
p |
|
CMS-1 |
12/62 (19%) |
4/28 (14%) |
0.1655 |
|
CMS-2 |
23/62 (37%) |
6/28 (21%) |
|
|
CMS-3 |
13/62 (21%) |
12/28 (43%) |
|
|
CMS-4 |
14/62 (23%) |
6/28 (21%) |
|
|
|
Cyclin D1 > median |
Cyclin D1 <= median |
p |
|
CMS-1 |
6/44 (14%) |
10/46 (22%) |
0.7457 |
|
CMS-2 |
15/44 (34%) |
14/46 (30%) |
|
|
CMS-3 |
12/44 (27%) |
13/46 (28%) |
|
|
CMS-4 |
11/44 (25%) |
9/46 (20%) |
|
Comment-14: Results, lines 227-228: the explanation of “TCF12-MALAT1 alliance” could be mentioned earlier in the text, or at least repeated.
[RESPONSE]: We have mentioned “TCF12-MALAT1 alliance” earlier in the Introduction section (lines 98-101) of the revised manuscript.
Comment-15: Results, line 236: why there are “TCF12” and “MALAT1” twice in Figure 1D? This is crucial since their effect seems to be different. In Figure 1E, “TCF12” and “MALAT1” are shown once, similar to their alliance.
[RESPONSE]: To comprehensively evaluate the contribution of the TCF12-MALAT1 alliance to CRC prognosis, we conducted Cox regression analyses on the interaction term of TCF12 mRNA and MALAT1 expressions, referred to as “TCF12´MALAT1”. In Figure 1D, we performed univariate Cox regression analyses on variates including gender, age, metastasis, TCF12 mRNA expression level, MALAT1 expression level, and TCF12´MALAT1. The evaluation of TCF12×MALAT1 involves more than simply multiplying their expression levels; it also requires adjusting with their individual effects. As such, “TCF12´MALAT1” was represented as three covariates in a multivariate Cox regression analysis: TCF12 mRNA expression level, MALAT1 expression level, and the multiplication of TCF12 mRNA and MALAT1 expression levels. In Figure 1E, we performed a multivariates Cox regression analysis on covariates including age, metastasis, and TCF12´MALAT1. We have amended Figure 1D and 1E a bit to render them more comprehensible, and added extra statements in the 2.6. section (lines 198-204), 3.1. section (lines 239-244), and Figure 1D legend (lines 268-273) of the revised manuscript.
Comment-16: Results, lines 251-253: contact the journal’s Office if significant p-values could be marked with red font or maybe it would be better to make it bold and with normal black font. Alternatively, there could be a normal black font and significance marks next to the value. The same could be done in Figures.
[RESPONSE]: We have used bold black font instead of red font in updated Table 1 and Table 2.
Comment-17: Results, lines 260-261: Figure 2A should be visualized as a Venn diagram or upset-plot. Currently, it is less scientific. Moreover, from my understanding – TCF12 targets are on the DNA level whereas those of MALAT1 are on the RNA level? This must be described in the text. Did take into account the possibility of creating an RNA-DNA hybrid in the case of MALAT1? This would allow you to indicate an even more important set of genes which are regulated by MALAT1 on both DNA and RNA levels.
[RESPONSE]: We have used a Venn diagram instead of old one. My lab has been doing about MALAT1’s function as a guide RNA to form triplex structures with several gene promoter regions. We have added some statements in the 2nd paragraph of the Discussion section (lines 470-506).
Comment-18: Results, lines 263-264: are all pathways significant in Figure 2B? Is 0.5 value a threshold of significance?
[RESPONSE]: Yes, all pathways are significant. In MGSA, a posterior probability (PP) threshold of 0.5 is typically suggested as a default criterion to determine if a pathway is considered “significant”. This means that pathways with PP > or = 0.5 are regarded as likely to be active or enriched.
Comment-19: Results, Figure 2C: are colors necessary? I think it would be much more valuable to color only significant results correlations and use a background color of either red (for positive correlation) or blue (for negative correlation). This would be much more self-explanatory while not overwhelming.
[RESPONSE]: A new Figure 2C has been used instead.
Comment-20: Results, lines 278-282: information about acquiring CMS classification should be mentioned in the methodology. Ideally, you should explain all subtypes in Methods whereas in Results there should be only a percentage of each subtype without its molecular description.
[RESPONSE]: As suggested, please see the revised 2.2. section (lines 130-134) and revised 3.2. section (lines 308-310).
Comment-21: Results, line 295: pairwise p-values would be much more valuable, because probably CMS1 vs CMS2 or CMS4 comparison is significant.
[RESPONSE]: The data are shown below:
|
|
Coef |
Hazard ratio |
p |
|
CMS-1 |
Reference |
|
|
|
CMS-2 |
-0.60 |
0.55 |
0.177 |
|
CMS-3 |
-0.52 |
0.60 |
0.328 |
|
CMS-4 |
-0.66 |
0.52 |
0.165 |
Comment-22: Results, lines 341-342: how were tumor recurrence and metastasis included if the only endpoint was overall survival? Or maybe they were assessed on the basis of other data than survival?
[RESPONSE]: Chi-square and KM (with log rank) were used, respectively. To avoid misleading, we have separated the original statement into two sentences. Please see the revised line 369 and line 376.
Comment-23: Results, line 361: you can mention that “TMBC” (or “TMCC”, if you address my first comment) comes from the first letter or each gene symbol.
[RESPONSE]: Same as the RESPONSE to Comment-1.
Comment-24: Discussion, line 442: delete “long” before “lncRNA”, because the meaning is already in the abbreviation.
[RESPONSE]: Corrected as indicated (line 490).
Comment-25: Discussion, lines 483-484: is this possible to suggest what could be the primary risk factor if not metastasis? Given the biological functions of gene products of TCF12/MALAT1/b-catenin/cyclinD1?
[RESPONSE]: We have added a new Figure 4F. Extra statements have been included into the revised Results section (lines 407-411), revised Discussion section (line 531-533), and Figure 4 legend (lines 448-451).
Comment-26: Discussion, lines 485-513: I think that the description of Wnt signaling is too long and deviates from the topic. Please reduce it a bit.
[RESPONSE]: We have reduced the length of the 4th paragraph of the Discussion section.
Comment-27: The “Conclusion” section is missing while I think the manuscript would benefit from the simple take-home message from your study.
[RESPONSE]: We have added an extra paragraph on the bottom of the Discussion section to discuss the limitations of our present study.

Reviewer 3 Report
Comments and Suggestions for Authors
Major point;
Authors should show what factors were analyzed in the multivariate analysis in Fig 4D. In lines 373-374, authors state that 'TMBC pattern is a metastasis-independent and even stronger prognostic factor for poorer overall survival in CRC patients', but to do so, authors should analyze Stage I/II/III cases and Stage IV cases separately and show that TMBC pattern is a prognostic factor.
Author Response
Authors should show what factors were analyzed in the multivariate analysis in Fig 4D. In lines 373-374, authors state that 'TMBC pattern is a metastasis-independent and even stronger prognostic factor for poorer overall survival in CRC patients', but to do so, authors should analyze Stage I/II/III cases and Stage IV cases separately and show that TMBC pattern is a prognostic factor.
[RESPONSE]: We thank the valuable comment from the Reviewer #3. We have added the data as Supplemental Figure 3, and we also include extra statements to make the content of Figure 4 more understandable (Ines 389-402).

Round 2
Reviewer 2 Report
Comments and Suggestions for Authors
Dear Authors, thank you for addressing my comments duly. I endorse the publication of your work. Just a small suggestion, regarding my last comment from the first round - I appreciate the addition of study limitations. Still, I think a separate Conclusions section should improve your manuscript even more. Other than that - congratulations!
Author Response
Comment: “Dear Authors, thank you for addressing my comments duly. I endorse the publication of your work. Just a small suggestion, regarding my last comment from the first round - I appreciate the addition of study limitations. Still, I think a separate Conclusions section should improve your manuscript even more. Other than that - congratulations!”
[Response]: We have added a “5. Conclusion” section into the revised manuscript (lines 606-619).

Reviewer 3 Report
Comments and Suggestions for Authors
Only Stage IV is metastatic. Stages I, II, and III are non-metastatic or resectable.
Author Response
Comment: “Only Stage IV is metastatic. Stages I, II, and III are non-metastatic or resectable.”
[Response]: Regarding the definition and criteria of colorectal cancer TNM staging (the American Joint Committee on Cancer TNM system), please refer to the link:
https://www.cancer.org/cancer/types/colon-rectal-cancer/detection-diagnosis-staging/staged.html
In brief, the TNM-III CRC patients have lymph node metastasis and the TNM-IV CRC patients exhibit distant metastasis. Therefore, we grouped the TNM-I & II patients as non-metastatic patients and the TNM-III & IV patients as metastatic patients for the further statistical analysis.
